# Hedging as Reward Augmentation in Probabilistic Graphical Models

**Debarun Bhattacharjya**
IBM Research
debarunb@us.ibm.com

**Radu Marinescu**
IBM Research
radu.marinescu@ie.ibm.com

## Abstract

Most people associate the term 'hedging' exclusively with financial applications, particularly the use of financial derivatives. We argue that hedging is an activity that human and machine agents should engage in more broadly, even when the agent's value is not necessarily in monetary units. In this paper, we propose a decision-theoretic view of hedging based on augmenting a probabilistic graphical model – specifically a Bayesian network or an influence diagram – with a reward. Hedging is therefore posed as a particular kind of graph manipulation, and can be viewed as analogous to control/intervention and information gathering related analysis. Effective hedging occurs when a risk-averse agent finds opportunity to balance uncertain rewards in their current situation. We illustrate the concepts with examples and counter-examples, and conduct experiments to demonstrate the properties and applicability of the proposed computational tools that enable agents to proactively identify potential hedging opportunities in real-world situations.

## 1 Introduction

Hedging is mostly synonymous with the financial industry. Wikipedia defines a (financial) hedge as "an investment position intended to offset potential losses or gains that may be incurred by a companion investment." Various relevant financial instruments are noted, including stocks, exchange-traded funds, insurance, forward contracts, swaps, options, etc. The notion of a hedge is however also occasionally used in other contexts; for instance, a hedge in linguistics refers to a phrase that signifies caution or uncertainty, or has an intended effect of dampening the strength of a claim. The fundamental hedging idea that is so widely applied in finance can however be generalized. Informally, hedging can be viewed as minimizing the 'risk' that arises from uncertainty in existing potential gains/losses along some measure of value. It may be possible and indeed prudent for a risk-averse decision maker to try to balance the uncertainty in their existing situation in some fashion.

Behavioral experiments indicate that people are occasionally risk-seeking while making decisions, particularly when losses are involved [22]. However, such behavior is often associated with contextual cognitive biases that they are typically unaware of [26]; when they truly understand the implications of risk-seeking behavior and its meaning in utility theory as opposed to common parlance, most people are quick to reject it and adopt risk-averse behavior instead. Thus, from a prescriptive perspective, most real-world situations should involve risk-averse decision makers, and therefore there can be real practical benefits around investigating hedging opportunities. Indeed, we argue that hedging is an activity that human and machine agents should engage in more broadly, in the appropriate situations, regardless of whether the agent's value is in monetary units.

Probabilistic graphical models such as Bayesian networks [35] and influence diagrams [21] are effective at representing a wide variety of real-world uncertain situations – where the latter model explicitly represents decisions by a decision maker – and they are powerful modeling paradigms in practice since they can be learned through a combination of expert opinion and data [17]. In

36th Conference on Neural Information Processing Systems (NeurIPS 2022).

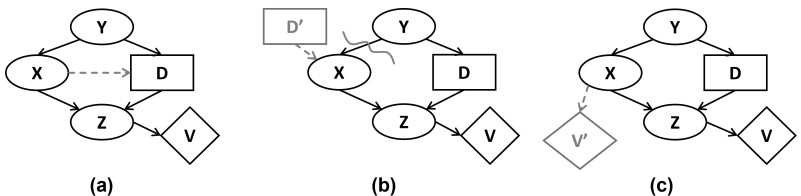

**(a)**  **(b)**  **(c)**

Figure 1: Graphical manipulations in an illustrative 5 node influence diagram, where $X$ is an uncertainty (random variable) of interest, $D$ is a decision and $V$ represents reward/value: (a) Information – $X$ observed before making decision $D$. (b) Control – Intervening on $X$ through decision $D'$. (c) Hedging – Augmenting reward $V'$ to $X$'s states.

this paper, we propose a general decision-theoretic approach and associated perspective for hedging related valuations, showcasing the required computations in real-world situations that are represented as probabilistic graphical models. The primary focus is around formalizing hedging as an analysis tool for probabilistic graphical models in general, where a decision maker can study potential benefits of hedging on uncertainties in their decision situation so as to explore opportunities for creating value by reducing their risk.

**Contributions.**    Our contributions are summarized as follows:

- We pose hedging as reward augmentation in a decision maker's situation as represented by a Bayesian network or influence diagram, and demonstrate how to evaluate hedging opportunities given a model. This view of hedging as graph manipulation is analogous to information gathering or intervening in a decision situation.
- We highlight a number of results around valuing hedging, using several examples to illustrate the concepts. Some results in particular could be helpful for agents to proactively seek out hedging as a way to improve their situation.
- We conduct experiments to demonstrate applications of hedging related computations as applied to probabilistic graphical models.

Note that in this work, the model is assumed to be known a-priori, possibly through a combination of statistical learning and/or causal discovery from data as well as assessments from domain experts and/or the decision maker(s). This makes the proposed analyses techniques indifferent to how models are learned and/or elicited.

## 2   Related Work

**Hedging in Finance.**    Since hedging is an activity that is centuries old, believed to have originated with a futures market for rice in Japan in the late 17th century, much has obviously been written about financial hedging. This includes literature around the extent to which hedging is used by corporations [15] and the degree to which corporate hedging is effective [33]. Instruments for hedging are studied in basic texts on finance [28], and machine learning algorithms are widely adopted by the financial industry in general, including for hedging [5, 6]. Although some of this literature in economics and finance is tangentially relevant to our work, it does not consider hedging in the context of probabilistic graphical models for representing uncertain situations.

**Graph Manipulations in Bayesian Networks & Influence Diagrams.**    The closest related work is around graph manipulations in Bayesian networks (both acausal and causal) and influence diagrams. In the influence diagram literature, this includes studying the benefits from information gathering, i.e. observing uncertainties before making a decision, and having the ability to control uncertainties and set them to desired states. The former manipulation is shown in the illustrative 5-node influence diagram in Figure 1(a), where an informational arc is added from uncertainty $X$ to decision $D$ (the uncertainty must not be downstream of the decision in general, to avoid cycles). This informs the *value of information* about $X$ prior to deciding [20]. There is significant literature on this subject, including around efficient computation [40, 24]. The latter manipulation, shown in Figure 1(b) where arcs from $X's$ parents are removed and augmented decision node $D'$ sets the state of $X$, is associated with the *value of control* [31, 27, 30, 41]. This is particularly popular in causal networks, forming the

basis for interventions in Pearl's do-calculus [36] as well as work that studies such causal semantics through a decision-theoretic lens [18, 8]. Hedging has been previously formulated as a buying price in prior work [39, 4]; here we expand upon this work by extending this notion to general value measures (not just monetary units), which vastly increases applicability of the concepts and tools in AI. Furthermore, we position hedging as a new kind of graph manipulation – augmenting a reward to some uncertainty $X$ – analogous to information and control, as illustrated in Figure 1(c) and described later in Section 4.

**Other Related Work (Reward Shaping, AI Safety, Robust ML, etc.)**  Since we frame hedging as reward augmentation, we briefly note some other related work, particularly around the implications of manipulating rewards as it pertains to reinforcement learning:

- There is a long line of research on reward shaping, which is a method where additional rewards are provided to guide a learning agent [10, 29, 34]. The goal in reward shaping is to make it easier to learn using supplemental rewards; in contrast, we frame hedging as an attempt to explore opportunities for minimizing risk as suitable for the decision maker in an already specified model.
- There may also be unintended and potentially harmful behavior emerging from poor design of real-world artificial intelligence systems, as explored in the literature on AI safety [1, 25]. One of the concerns is around reward hacking/tampering/gaming/corruption, such as where an agent finds a way to give itself high reward without completing its intended goals [2, 19]. There is ongoing work on methods to alleviate negative side effects of misspecified reward functions [16, 12].
- We note that hedging is pertinent for risk-averse decision makers and the subject of risk sensitivity has long been of interest in reinforcement learning [32, 37], with more recent application to problem settings where safe policies are desired [14].
- Hedging-like strategies can also be found in robust machine learning settings including decision-theoretic online learning [13].

## 3   Value, Utility and Graphical Models

We use the framework of probabilistic graphical models – in particular, Bayesian networks and influence diagrams. Since the notions of value/reward and utility are crucial to define hedging, we begin with some relevant background.

### 3.1   Value/Reward and Utility Functions

Preference modeling is widely studied in AI, economics, management science, behavioral decision making, marketing and related fields. In this paper, we follow a preference modeling approach from decision analysis that has also been used for valuing information [20, 40, 4]. Specifically, we assume the decision maker's preferences are decomposed into a *measurable value function*[1] $v(\cdot)$ [11, 23] over domain $\mathbb{R}$, as well as a one-dimensional utility function $u(\cdot) : \mathbb{R} \to \mathbb{R}$ that is monotonic over the value measure [43]. $v(\cdot)$ is said to be 'riskless' because it captures strength of preference for situations with no uncertainty, whereas $u(\cdot)$ captures the decision maker's attitude towards risk, i.e. lotteries that involve uncertainty in the value measure.

The main motivation for using measurable value functions is that it enables one to work with concepts that involve magnitudes. For example, an ordinal measure which only provides rankings and no other quantitative implications may be insufficient for modeling measures of interest to a decision maker, such as monetary units, lives saved, number of customers, quality-adjusted life years, lead time, etc. Crucially for our work, the preference decomposition ensures hedging valuations are in units that are meaningful to the decision maker.

The decision maker's *risk aversion* coefficient is the ratio of the second derivative and first derivative of the utility function $u(\cdot)$: $\gamma(v) = -\frac{u''(v)}{u'(v)}$ [3]. The concavity (convexity) of the utility function provides an indication of the decision maker's risk aversion (risk seeking behavior). Some parametric forms for utility functions are popular in practice due to their computational properties. The (monotonically increasing) exponential utility function follows $a - b \, \text{sgn}(\rho) \exp{(-x/\rho)}$, where $a$ and $b > 0$ are

---

[1]Readers familiar with reinforcement learning (RL) should view this as a 'reward function' and not what is typically know as 'value function' in RL. Here we use these terms interchangeably and often use them both together through the term 'value/reward' as a means of clarification.

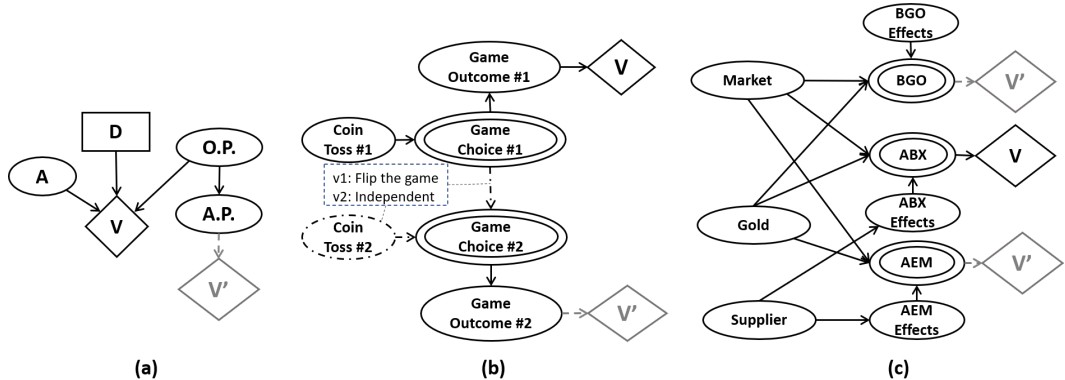

Figure 2: Examples of Bayesian networks (BNs) and influence diagrams (IDs): (a) Oil wildcatter ID with a potential augmented reward on airline price (A.P.) [39]. (b) BN for (two versions of) the Game example. v1: Flipped second game; v2: An independent coin toss for second game. Hedging is explored on Game Outcome #2. (c) Gold mining BN with two potential augmented rewards [42].

constants and sgn($\rho$) is the sign of the *risk tolerance* $\rho \neq \infty$. The power utility function is of the form $a + b\,\text{sgn}(\alpha)\,\text{sgn}(x)\,|x|^\alpha$, with constants $a$ and $b > 0$, where $\text{sgn}(\alpha)$ and $\text{sgn}(x)$ are the signs of $\alpha \neq 0$ and $x$ [38]. The power function exhibits discontinuous behavior around 0 for $\alpha < 0$ [44]. The limiting cases for the exponential and power functions are the linear and logarithmic functions, respectively. Many of our results apply only for linear and exponential utility functions, because they are the only ones that satisfy constant risk aversion, i.e. where $\gamma(v)$ is a constant [3]. Note that a linear utility function signifies *risk neutrality*, which is the most widely used assumption in many applications (sometimes implicitly), for instance in most (but not all) models for reinforcement learning. Exponential utility functions are also popular; the risk tolerance $\rho$ in these utility functions is the reciprocal of the (constant) risk aversion coefficient $\gamma$.

## 3.2 Bayesian Networks and Influence Diagrams

A **Bayesian network** (henceforth BN) $\mathcal{B}$ represents probabilistic dependencies between random variables through a directed acyclic graph. It consists of a graph $\mathcal{G}$ with a node for each uncertain variable, along with a conditional probability distribution (CPD) for each node. We denote uncertain nodes as $\boldsymbol{\mathcal{X}} = \{X_1, \cdots, X_N\}$, such that each $X_i$ has domain $Dom(X_i)$ and is associated with CPD $P(X_i|\boldsymbol{Pa(X_i)})$ where $\boldsymbol{Pa(X_i)}$ are the parents of $X_i$ in $\mathcal{G}$.

While it is not traditional to include additional types of nodes in a BN, we also include value/reward nodes since hedging requires value. Thus, we assume that the BN also includes value nodes $\boldsymbol{\mathcal{V}} = \{V_1, \cdots, V_K\}$ which are sinks and only have parents from $\boldsymbol{\mathcal{X}}$. Each value node $V_k$ is assumed to be a scalar in $\mathbb{R}$, and is associated with a value/reward function $v_k(\cdot) : Dom(\boldsymbol{Pa(V_k)}) \to \mathbb{R}$. The total value is an aggregated function over the individual value nodes, $V = a(\boldsymbol{\mathcal{V}})$. Summation and multiplication are common modes of aggregation. Most of the results presented here apply for additive value aggregation. In many applications (including but not limited to finance), $V$ is in monetary units, which is a natural measure by which many real-world decision situations are valued, but other measures are also possible and indeed used in examples in this paper.

Even though there are no explicit decisions for the decision maker in this Bayesian network model augmented with value nodes, we still refer to their situation as a decision situation. This is both for simplicity but also in recognition of the fact that there could be potential future opportunities that the decision maker may wish to include explicitly as decisions.

An **influence diagram** (henceforth ID) $\mathcal{I}$ is a graphical model of a decision maker's decision situation under uncertainty and can be viewed as an extension and generalization of a BN. It can represent sequential decision situations such as Markov decision processes (MDPs) and partially observable MDPs (POMDPs). Along with uncertainties $\boldsymbol{\mathcal{X}}$ and value nodes $\boldsymbol{\mathcal{V}}$, they include decisions $\boldsymbol{\mathcal{D}} = \{D_1, \cdots, D_M\}$. The domain of a decision $Dom(D_i)$ is the set of alternatives (also known as actions) that the decision maker can take. Arcs into decision nodes have special semantics – they

are informational arcs, indicating that the parents of the decision have been observed before it is made. Arcs into other nodes are probabilistic or deterministic, like in BNs. We make the standard assumptions: a *total ordering* of decisions as well as *no forgetting*, i.e. parents for a decision are also parents of subsequent decisions. Figure 2 shows examples of BNs and IDs, with decisions (rectangles), uncertainties (ovals – concentric if deterministic) and values/rewards (diamonds).

Similar to what was described for BNs, the value nodes of an ID $\mathcal{I}$ are aggregated into total value, $V = a(\mathcal{V})$. The decision maker's preferences under risk are modeled using one-dimensional utility function $u(\cdot)$, which is applied on measurable value $V$ with domain $\mathbb{R}$ and is denoted by $f(\mathcal{V}) = u(a(\mathcal{V}))$. A policy for $\mathcal{I}$ is a list of decision rules $\Pi = (\Pi_1, \ldots, \Pi_M)$ where each rule $\Pi_i$ is associated with a decision $D_i$ and is a mapping from the domain of its parents to an alternative in the domain of $D_i$, $\Pi_i = d(\boldsymbol{Pa(D_i)})$, such that $d(\cdot) : Dom(\boldsymbol{Pa(D_i)}) \rightarrow Dom(D_i)$. The optimal policy of $\mathcal{I}$ maximizes the expected utility, $\Pi^* = \mathrm{argmax}_\Pi E_{\mathcal{X},\Pi}[f(\mathcal{V})]$. An ID at a fixed policy can be drawn in policy form, also known as a *policy diagram* [7], where decisions have a deterministic or probabilistic relationship with its parents, thus effectively converting the ID to a BN.

## 4 Hedging as Reward Augmentation

We describe a general notion of hedging, posed as reward augmentation in a probabilistic model. We will generally refer to the model as an ID, calling out situations where there are no decision nodes.

Consider an agent whose decision situation is represented by an ID $\mathcal{I}$ with uncertain nodes $\mathcal{X}$, value nodes $\mathcal{V}$, and optionally decision nodes $\mathcal{D}$. When there are no decision nodes, this is a BN $\mathcal{B}$. Since an ID in policy form is equivalent to a BN, all its corresponding results apply to BNs as well. The agent's *certain equivalent* (CE) for their (decision) situation (represented by the ID/BN) is the certain amount, in units of value $V$, at which they are indifferent between this amount and their situation. For instance, if the agent has utility function $u(\cdot)$ and faces two potential outcomes of value $v1$ (with probability $p$) and $v2$ (with probability $1 - p$) respectively, then the following must hold: $u(CE) = p * u(v1) + (1 - p) * u(v2)$, implying that $CE = u^{-1}(p * u(v1) + (1 - p) * u(v2))$.

We first define a decision maker's *prior* certain equivalent, i.e. without any augmentation to their prior situation:

**Definition 1.** *For an agent with BN $\mathcal{B}$, their prior certain equivalent $CE_0 = u^{-1}(E_{\mathcal{X}}[f(\mathcal{V})])$.*

**Definition 2.** *For an agent with ID $\mathcal{I}$, their prior certain equivalent at policy $\Pi$ is $CE_0^\Pi = u^{-1}(E_{\mathcal{X},\Pi}[f(\mathcal{V})])$. Their (optimal) prior certain equivalent is $CE_0^* = u^{-1}(\max_\Pi E_{\mathcal{X},\Pi}[f(\mathcal{V})])$.*

The key difference between the two CE expressions for IDs is that one is at fixed policy $\Pi$ whereas the other involves an optimization over policies. We now formalize a hedge as reward augmentation:

**Definition 3.** *A **hedge** in a BN or ID is an augmented reward $r'(\cdot)$ for uncertainties $\mathcal{X_S} \subset \mathcal{X}$. This is a deterministic function that introduces a new value node $V'$ with parents $\mathcal{X_S}$ in the agent's situation, s.t. $V' = r'(\mathcal{X_S})$. A hedge is **simple** if the associated uncertainty set $\mathcal{X_S}$ is a single node in the graph. It is **balanced** for additive value aggregation if $E[r'(\mathcal{X_S})] = 0$, and for multiplicative value aggregation if $E[r'(\mathcal{X_S})] = 1$.*

**Example 1** (**Oil Wildcatter**). Consider an oil wildcatter whose decision situation is represented by the ID in Figure 2(a) [39]. Their profit depends on their drilling decision (D), oil price (O.P.) and amount of oil (A) in the reservoir. The wildcatter has also included an uncertain node for airline stock price (A.P.), since it is negatively correlated with the price of crude oil. Augmenting their situation with a reward that is a deterministic function of any of the three uncertain nodes would result in a simple hedge. Hedging on node A.P. is a classic case of corporate financial hedging. □

### 4.1 The Value of a Hedge

There is a natural important question to address: how much does a hedge balance out the uncertainty in the rewards from the current situation, for a decision-theoretic decision maker? The hedging value of the augmented reward is defined as the additional improvement in the agent's certain equivalent from including it, beyond the certain equivalent of the augmented reward alone. We distinguish between the case for when an ID is fixed at a policy $\Pi$ vs. when re-optimization is allowed, post reward augmentation. The former case is identical to that for a BN. Formally, they are defined as:

**Definition 4.** *The value of hedging of augmented reward $r'(\cdot)$ over nodes $\boldsymbol{\mathcal{X}_S}$ at policy $\Pi$ for additive value aggregation is:*

$$VoH^{\Pi}(r', \boldsymbol{\mathcal{X}_S}) = u^{-1}(E_{\boldsymbol{\mathcal{X}},\Pi}[f(\boldsymbol{\mathcal{V}}, V')]) - CE_0^{\Pi} - u^{-1}(E_{\boldsymbol{\mathcal{X}},\Pi}[f(V')]), \qquad (1)$$

*where $V'$ is the new value node from the augmented reward and $CE_0^{\Pi} = u^{-1}(E_{\boldsymbol{\mathcal{X}},\Pi}[f(\boldsymbol{\mathcal{V}})])$.*

**Definition 5.** *The value of hedging of augmented reward $r'(\cdot)$ over nodes $\boldsymbol{\mathcal{X}_S}$ for additive value aggregation is:*

$$VoH^{*}(r', \boldsymbol{\mathcal{X}_S}) = u^{-1}(\max_{\Pi} E_{\boldsymbol{\mathcal{X}},\Pi}[f(\boldsymbol{\mathcal{V}}, V')]) - CE_0^{*} - u^{-1}(\max_{\Pi} E_{\boldsymbol{\mathcal{X}},\Pi}[f(V')]), \qquad (2)$$

*where $V'$ is the new value node from the augmented reward and $CE_0^{*} = u^{-1}(\max_{\Pi} E_{\boldsymbol{\mathcal{X}},\boldsymbol{\Pi}}[f(\boldsymbol{\mathcal{V}})]$.*

**Remark.** The reason for including the third term in both $VoH$ definitions is to distinguish the value of the *hedge* from the incremental value of the reward augmentation alone, which would only include the first two terms. This can help reveal the specific effect that probabilistic dependence between rewards has on the value to the decision maker, beyond the value from the augmented reward by itself. When only the first two terms are included, we refer to the computation as value of augmented reward ($VoAR$). Also, for multiplicative value aggregation, we replace subtraction with division. Note that $VoH$ can be positive or negative or zero.

We illustrate the value of hedging with the following examples. For a BN, both versions of $VoH$ ($VoH^{\Pi}$ and $VoH^{*}$) are identical since there is no policy to optimize.

**Example 2** (**The Game**). Consider a deal between two players who will toss a coin to determine a game to play: either fencing or golf. The loser must pay \$1000 to the winner. Suppose both players believe that the probabilities of player#1 winning at fencing and golf are 0.9 and 0.3 respectively. Suppose also that both players are risk-averse with exponential utility functions, with the same risk tolerance $\rho = \$3000$. The players' (prior) certain equivalents are \$35.72 and $-\$350.63$ respectively. Note that if the players were risk-neutral, their certain equivalents would be \$200 and $-\$200$ respectively; since they are risk-averse, their $CE$s are lower. Computational details for all examples are provided in Appendix B.

Figure 2(b) (v1) models this situation from either player's perspective as a BN, where concentric ovals indicate a deterministic relation. This figure also includes the possibility of playing a *second* game of the other type. Thus, players would play *both* types of games regardless of the coin flip outcome. What is the value of hedging of the second game outcome to both players, where the payoffs are the same as the first one? For player#1, $VoH$ can be computed as \$130.3; for player#2, this is \$100.22 (see Appendix B.1). $VoH$ is positive for both players in this example, implying that the dependence between the games is helping balance the uncertainty for both. $\qquad \square$

The value of hedging at a fixed policy has the following desirable properties for utility functions that satisfy constant risk aversion (linear and exponential):

**Theorem 6.** *For additive value function aggregation:*

  *i  For a risk-neutral decision maker, $VoH^{\Pi}$ for any augmented reward is $0$.*
  *ii  For an exponential utility function, if $X_S$ and value nodes $\boldsymbol{\mathcal{V}}$ are independent under policy $\Pi$, $VoH^{\Pi}$ for any augmented reward is $0$.*

**Example 3** (**The Game**). Consider Figure 2(b) (v2) where the players are considering playing a second game, but this time they will flip the coin again, therefore the arc between game choices is absent. Here the value of hedging of the second game outcome is $0$ for both players, ex: for player#2, this is $\$(-701.26 + 350.63 + 350.63) = \$0$ (see Appendix B.1). Given that player#2 has already struck a deal to play the first game, they would rather not play again in either of the versions in Figure 2(b) – but if they had to choose, they would choose the hedge from v1 and they would be willing to pay up to the $VoH$ of \$100.22 ($-701.26$ vs. $-601.04$). This is the hedging benefit to the player – they would rather play both games than their weaker game (fencing) twice! $\qquad \square$

The afore-mentioned results around zero hedging value for a risk-neutral decision maker or for uncertainties that are irrelevant to value are perhaps consistent with the reader's intuitive understanding for how hedging should work. However, these results do not extend in general to other cases, such as for other utility functions, as the next example illustrates, or when the policy can be re-optimized.

**Example 4** (**The Game**). We revisit the version of the game in Figure 2(b) with a second independent coin toss (v2). Suppose player #1 has a logarithmic utility function of the form $u(v) = log(v + w)$, where $w$ is their initial wealth, assumed to be \$5000. Here $VoH = \$(283.18 - 101.7 - 101.7) = \$79.78$ (see Appendix B.1). We consider this to be a counter-example: even though the second coin toss is independent from the first and associated uncertain rewards, there is still value from hedging due to 'wealth effects' arising from utility functions that do not satisfy constant risk aversion. □

## 4.2 The Value of a Perfect Hedge

Thus far, we have described situations where the agent is considering hedges where the rewards based on some uncertainties are known a-priori. While such potential rewards may be known in some applications, there are other situations where this may not be the case. How can an agent with a graphical model to represent their decision situation pro-actively seek out hedging opportunities without knowing the actual rewards? The notion of a *perfect hedge* is formalized, again both at a fixed policy and when re-optimization is allowed.

**Definition 7.** *The value of a perfect hedge on uncertainty $X$ at policy $\Pi$ for additive value aggregation is the simple and balanced augmented reward on $X$ that optimizes the increase in certain equivalent at policy $\Pi$, with and w/o the hedge:*

$$VoPH^{\Pi}(X) = \max_{r'(\cdot) \in \mathcal{R}(X)} \left( u^{-1}(E_{\boldsymbol{\mathcal{X}},\Pi}[f(\boldsymbol{\mathcal{V}}, V')]) \right) - CE_0^{\Pi}, \tag{3}$$

*where $V'$ is the new value node from the balanced hedge, $\mathcal{R}(X)$ denotes the set of balanced rewards for $X$, and $CE_0^{\Pi} = u^{-1}(E_{\boldsymbol{\mathcal{X}},\Pi}[f(\boldsymbol{\mathcal{V}})])$.*

**Definition 8.** *The value of a perfect hedge for an uncertainty $X$ that is not downstream of any decision, for additive value aggregation, is the simple and balanced augmented reward on $X$ that optimizes the increase in certain equivalent, with and w/o the hedge:*

$$VoPH^*(X) = \max_{r'(\cdot) \in \mathcal{R}(X)} \left( u^{-1}(\max_{\Pi} E_{\boldsymbol{\mathcal{X}},\Pi}[f(\boldsymbol{\mathcal{V}}, V')]) \right) - CE_0^*, \tag{4}$$

*where $V'$ is the new value node from the balanced hedge, $\mathcal{R}(X)$ denotes the set of balanced rewards for $X$, and $CE_0^* = u^{-1}(\max_{\Pi} E_{\boldsymbol{\mathcal{X}},\boldsymbol{\Pi}}[f(\boldsymbol{\mathcal{V}})]$.*

Recall that a balanced reward for additive value aggregation is one whose expected value is zero. Thus, for additive aggregation, the perfect hedge is the one with zero expected reward that maximizes the certain equivalent, both at policy and otherwise, given this hedge.

**Remark.** A perfect hedge is intended for when the agent is trying to identify which uncertainties in their situation are most amenable for hedging. We make a few observations: 1) Unlike in the definition(s) for $VoH$, $VoPH$ does not depend on $r(\cdot)$ by design, since it is unknown; 2) The third term from the $VoH$ definition(s) is discarded here since the focus is on incremental value from the best possible balanced hedge; 3) When re-optimization is allowed, it may be possible for a decision to have an impact on a downstream uncertainty, i.e. different policies may change the distribution of the augmented reward $V'$. Thus we add the caveat for $VoPH^*$ to disallow any invalid uncertainty.

The following intuitive results around $VoPH$ hold for some important and practical special cases:

**Theorem 9.** *For additive value function aggregation:*

 i *For a risk-neutral decision maker and any valid uncertainty $X$, $VoPH^*(X) = 0$ and $VoPH^{\Pi}(X) = 0$ for any policy $\Pi$.*
 ii *For any valid uncertainty $X$, $VoPH^*(X) \geq 0$ and $VoPH^{\Pi}(X) \geq 0$, $\forall \Pi$.*
 iii *For a risk-averse decision maker with an exponential utility function, and any uncertainty $X$, the value of any hedge $r(\cdot)$ on $X$ at policy $\Pi$ is bounded above by $VoPH^{\pi}(X) + E_{\boldsymbol{\mathcal{X}},\Pi}[V']$, where $V'$ is the new value node from the hedge.*

The last result above is important as it can help the agent reject any potential hedges based on $VoPH$. This is analogous to value of information analysis where some information gathering sources can be rejected immediately if they cost more than the value of perfect information, i.e. the additional value to a decision maker when an uncertain variable is observed before making a decision [20].

**Example 5** (**Oil Wildcatter**). Let us revisit the oil wildcatter example in Figure 2(a) with model parameters as provided in Appendix B.2 [39]. With these numbers, it is optimal for the oil wildcatter to drill a-priori, with a certain equivalent of \$38.99M. If we fix the policy at drill (d), the value of perfect hedging for each uncertainty can be computed: $VoPH^d(A) = \$7.68$M, $VoPH^d(OP) = \$3.26$M, $VoPH^d(AP) = \$1.61$M. Hedging on the amount of oil seems comparatively most lucrative, which might be realizable by finding a dependent reservoir nearby, preferably one with a common source with the current reservoir. Hedging on airline price could occur through purchase of airline stocks.

$VoPH^d$ on airline price is lower than that on oil price, since the latter is directly involved in the determination of the monetary prospects. Even though the dependence between these uncertainties is very high, there is a marked difference in potential value from hedging – $VoPH$ quantifies these differences. According to Theorem 9(iii), the wildcatter should not be willing to pay more than $VoPH^d(AP) = \$1.61$M (once they have decided to drill) for *any* deal whose prospects are determined solely by the states of $AP$ (high and low) and whose mean equals $0$. Lower $VoPH$ means that the decision maker can eliminate more potential deals before getting into specifics.  □

It may seem intractable to compute $VoPH$, based on its definition(s), which involves an optimization over rewards. However, $VoPH^\Pi$ for a fixed policy is a difference of expectations which can be computed efficiently [4].

**Theorem 10.** *For additive value function, an exponential utility function, and any uncertainty $X$, the perfect hedge (augmented reward) and $VoPH$ for $X$ at policy $\Pi$ are:*

$$r^*(X) = E_X(CE^\Pi|X) - CE^\Pi(x); \ VoPH^\Pi(X) = E_X(CE^\Pi|X) - CE_0^\Pi,$$

*where $CE^\Pi(x)$ is the certain equivalent given state $x$ of $X$ is observed, $E_X(CE^\Pi|X)$ is its expectation over all states of $X$, and $CE_0^\Pi$ is the prior certain equivalent at policy $\Pi$.*

**Remark.** The intuition behind the above result is described in the proof sketch in Appendix A, but importantly it shows that $VoPH^\Pi(X)$ can be computed efficiently for any uncertain node $X$. By converting the ID to policy form, one can apply standard BN inference algorithms to compute the required certain equivalents after setting evidence for $X$ at its various states. Computing $VoPH^*$ is a harder problem though, deserving further future attention.

## 5 Experiments

Hedging related computations in graphical models are holistic in that they leverage the entire model, including the utility function, thus they do not necessarily conform to measures that solely rely on dependence/correlation to identify hedging opportunities. This is where concepts such as $VoH$ and $VoPH$ are useful in practice, as they account for all modeling aspects together. We conduct experiments with both synthetic and real data to demonstrate applicability of hedging valuations as well as some complexities that are non-trivial and potentially counter-intuitive. Since our contributions in this paper are primarily conceptual and apply to any BN or ID, we note that the required computations, as formalized by equations in Definitions 4, 5, 7 and 8, can be performed using any of the existing state-of-the-art BN inference methods and ID solvers.

### 5.1 Auto ML

Figure 3 shows an ID where an automated machine learning agent must choose a setting (S) which affects its uncertain performance on different tasks. For instance, this could be an auto AI pipeline decision for some future tasks. Value is measured by a multiplicative positive metric such as % performance improvement compared to a baseline algorithm. The agent is assumed to be risk-averse with utility function $u(v) = log(v)$. The agent is currently committed to two tasks (P1 and P2) but is considering an opportunity to hedge on a third task (P3). Since tasks are conditionally independent given the setting, one may not expect hedging to play a role.

The setting (S) is assumed to be binary with states high ($h$) and low ($l$). Each task $i$'s performance in setting $s$ is lognormally distributed with mean parameter $\mu_i^s$ and common standard deviation parameter $\sigma$. The expected performance for any task $\bar{P}_i^s$ is related to the mean and standard deviation parameter of the lognormal distribution as: $\bar{P}_i^s = e^{\mu_i^s + \sigma^2/2}$. We denote $\mu_i = [\mu_i^h, \mu_i^l]$ and $\bar{P}_i = [\bar{P}_i^h, \bar{P}_i^l]$ for compactness.

Consider the following numerical case: $\sigma = 1$, expected performances for tasks: $\bar{P}_1 = [2, 1]$, $\bar{P}_2 = [3, 2]$, $\bar{P}_3 = [1, 7]$. With these numbers, policy $\pi = h$ is optimal a-priori.

In this example, all computations can be done analytically due to the assumptions, and we obtain value of hedging $VoH^\pi = 1$, $VoH^* = 0.33$, and value of augmented reward $VoAR^\pi = 0.61$, $VoAR^* = 1.42$. Re-optimization changes $VoH$ and $VoAR$ in this case because the setting $l$ is so much better for P3 that it is chosen after reward augmentation. Note that fixing policy $\pi = h$ underestimates the actual value from augmentation, since the agent would prefer the $l$ setting if it could also accrue value from P3 ($VoAR$ is 0.61 vs. 1.42). Interestingly, although augmentation is desirable here again due to intrinsic value from P3, there is a reduction in value from hedging itself, since $VoH^* < 1$. (Since value is multiplicative here, $VoH = 1$ is equivalent to $VoH = 0$ for additive aggregation.) This example illustrates that allowing for decision re-optimization and a generic utility function besides the linear or exponential utility function such as logarithmic in this case can impact hedging valuations in unexpected ways.

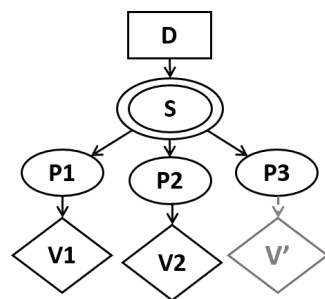

Figure 3: AutoML ID, with a potential augmented reward. $D$ is a decision, $S$ is the setting and $P1,P2,P3$ are performance tasks.

## 5.2 Gold Mining Stocks

Probabilistic graphical models are able to represent an uncertain situation by combining expert opinion and data. Figure 2(c) shows an example BN about three gold mining stocks: ABX, BGO and AEM [42]. Stock $j$'s monthly return is modeled through linear regression:

$$R_j = \beta_0 + \beta_M M + \beta_G G + \epsilon_j,$$

where $M$ and $G$ are the market and gold-industry specific indices, $\{\beta_0, \beta_M, \beta_G\}$ are regression coefficients, and $\epsilon_j$ is a Gaussian error term for stock-specific effects. ABX and AEM have a common supplier; the effect of the conditions at the supplier is modeled through the error terms. The numbers are based on real data from 1996 to 1998 (see Appendix C.1).

We assume the decision maker is only considering the gold mining industry, for whatever reason. They have committed to investing $1000 in ABX and are now considering a potential hedging opportunity by investing an additional $1000 in either BGO or AEM. Intuition suggests that BGO will be a relatively better hedge since ABX and AEM have a common supplier – Pearson correlation coefficients for (ABX, AEM) and (ABX, BGO) are 0.41 and 0.31 respectively.

Table 1 shows $VoH$ for varying risk tolerance. Unsurprisingly, all hedging valuations are negative since stocks belong to the same industry. However, AEM is the more appropriate choice between the two, mainly because of the higher variance in BGO returns. Note $VoH \to 0$ as risk tolerance increases. Computations for this example were done using Monte Carlo simulation with 10 runs of $100K$ samples each.

## 5.3 Supply Chain Risk

We consider a real-world supply chain risk BN model for a company's line of product manufactured in South-east Asia and shipped to USA [9]. The company is interested in managing the lead time in the supply chain. We simplified the original BN slightly, considering 18 uncertain nodes for different types of risks – including disasters, contractor related risks, customs, etc. – and 4 value nodes for lead time (in units of days) at sourcing, sub-assembly, transport and for other reasons due to disasters. Value is additive and $u(\cdot)$ is monotonically decreasing (less lead time is preferred), assumed to be exponential with risk tolerance $\rho$. The structure as well as all parameters of this BN are included in Appendix C.2.

Table 2 shows $VoPH$ computed for 7 key uncertainties for various levels of risk tolerance $\rho$, using the result in Theorem 10. Computations were done using exact Bayesian inference available from the pgmpy[2] package. The table reveals the most relevant sources of uncertainty for hedging purposes,

---

[2]https://github.com/pgmpy/pgmpy

Table 1: Value of hedging ($VoH$) on BGO and AEM stocks in dollars given the decision maker's current investment in ABX in the Gold Mining BN (Figure 2(c)) with varying risk tolerance ($\rho$). Monte Carlo error for computations is also indicated based on 10 runs of 100K samples each.

| Variable | $\rho = 100$ | $\rho = 200$ | $\rho = 500$ | $\rho = 1K$ |
|---|---|---|---|---|
| BGO | $-18.6$ $\pm 0.47$ | $-9.4$ $\pm 0.12$ | $-3.7$ $\pm 0.06$ | $-1.9$ $\pm 0.02$ |
| AEM | $-13.5$ $\pm 0.14$ | $-6.7$ $\pm 0.05$ | $-2.7$ $\pm 0.02$ | $-1.3$ $\pm 0.01$ |

Table 2: Value of perfect hedging ($VoPH$) in units of lead time (days) for 7 variables in the Supply Chain Risk BN (Figure 4, Appendix C.2) with varying risk tolerance ($\rho$). The two most valuable hedges are for natural disaster (ND) and geo-political instability (GI).

| Variable | $\rho = 10$ | $\rho = 50$ | $\rho = 100$ |
|---|---|---|---|
| EDI | 0.001 | 0 | 0 |
| CSP | 0.078 | 0.015 | 0.007 |
| CMB | 0.373 | 0.049 | 0.023 |
| DL | 0.003 | 0 | 0 |
| ND | 6.004 | 0.758 | 0.353 |
| LS | 0.033 | 0.007 | 0.003 |
| GI | 1.91 | 0.171 | 0.078 |

which in this case would primarily arise from insurance or contracts in practice. The company should clearly consider insurance against natural disasters (ND) and geo-political instability (GI) in particular, but there are other surprisingly important risks, such as contract manufacturer backruptcy (CMB) and to a lesser extent component sourcing problems (CSP). As the decision maker gets closer to being risk neutral, $VoPH \rightarrow 0$ for all uncertain nodes.

This application illustrates the primary purpose of $VoPH$ analysis: to identify potential hedging opportunities without necessarily having specific information about rewards and the cost of hedging at the time of the analysis. This is similar to pursuing information gathering opportunities before having details about the accuracy of such tests [20]. The actual valuation of any hedge here (such as insurance) would of course depend on the specifics of the reward, but $VoPH$ provides guidance around directions for the finance team to pursue, *before* they have identified specific hedges. Thus the goal is to provide initial support towards improving the decision situation by potentially reducing risk, similar to how value of information and control can provide guidance around potential information sources or intervening on uncertainties wherever possible. In general, $VoPH$ could be an important tool in an analyst's toolbox for improving their situation as represented by a BN or ID.

## 6   Conclusions

We have presented hedging as reward augmentation – a new kind of graph manipulation in BNs and IDs that represent a decision maker's situation, where preferences are represented by a one-dimensional utility function over a measurable value/reward. Most results apply for special cases such as additive value aggregation, and for decision makers with exponential utility functions. There are computational reasons why it is hard to generalize beyond a point, explaining why additive reward aggregation is pervasive in domains that compute optimal policies, but the fundamental ideas are more general and there may be opportunities to explore other cases for select problem classes.

There are open problems for future work, such as around computing the value of perfect hedging when re-optimization is allowed; since this is a difficult optimization problem, approximations may be necessary. We also stress that there may be risks of reward augmentation, for instance to AI safety. Side effects can be crucial for hedging, which if ignored during modeling could lead to undesirable effects. For instance, a company that is seen to be actively hedging may sacrifice shareholder confidence. Thus, while hedging may bring opportunities, there are potential risks around reward augmentation that should not be neglected. Hedging however also provides analysis opportunities for human and machine agents, as we have argued in this paper. Additional investigation of real-world problems is necessary to better understand potential benefits of hedging in AI systems and could lead to further methodological research progress.

## Acknowledgements

We thank Ross Shachter for several helpful discussions around decision-theoretic hedging as well as three anonymous reviewers for valuable feedback and suggestions.

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
