# A    Theorem Proofs

**Proof of Theorem 6(i)**:

*Proof.* The decision maker is risk-neutral, therefore we can choose $u(v) = v$; note that utility functions can be modified by linear transformations without changing the certain equivalent. This means that certain equivalents and expected values are equivalent. Therefore:

$$\begin{aligned}
VoH^{\Pi} &= E_{\boldsymbol{\mathcal{X}},\Pi}[f(\boldsymbol{\mathcal{V}}, V')] - E_{\boldsymbol{\mathcal{X}},\Pi}[f(\boldsymbol{\mathcal{V}})] - E_{\boldsymbol{\mathcal{X}},\Pi}[V'] \\
&= E_{\boldsymbol{\mathcal{X}},\Pi}[f(\boldsymbol{\mathcal{V}})] + E_{\boldsymbol{\mathcal{X}},\Pi}[V'] \\
&\quad - E_{\boldsymbol{\mathcal{X}},\Pi}[f(\boldsymbol{\mathcal{V}})] - E_{\boldsymbol{\mathcal{X}},\Pi}[V'] \\
&= 0
\end{aligned}$$

The last but one equation arises due to additive value aggregation and the linear utility function.    □

**Proof of Theorem 6(ii)**:

*Proof.* We drop the policy superscript in all relevant expressions for notational simplicity. We use a property of the exponential utility function around linear tranformation: if every reward is augmented by a constant amount, then the certain equivalent increases by that constant amount. In general, this holds only for linear and exponential utility functions as they are the only ones satisfying constant risk aversion.

Consider the last term of $VoH$, where there is reward only from the augmented reward node $V'$. Let us denote this certain equivalent as $\hat{y} = u^{-1}(E_{\boldsymbol{\mathcal{X}},\Pi}[f(V')])$. Now consider the situation where reward $\hat{y}$ is added to every outcome in the prior situation. We refer to its certain equivalent as $CE^{\dagger}$. From the afore-mentioned property of exponential utility functions, $CE^{\dagger} = CE_0 + u^{-1}(E_{\boldsymbol{\mathcal{X}},\Pi}[f(V')])$.

The required result holds if the certain equivalent of the situation which includes the augmented reward, denoted $CE'$, is also $CE^{\dagger}$. We show this by also using independence between this new reward and existing rewards. Suppose the existing reward outcomes are denoted $x_i$ with probabilities $p_i$, and the new reward has outcomes $y_j$ with probabilities $q_j$. Then, the expected utility $EU'$ is the probability weighted average $\sum_i p_i \sum_j q_j u(x_i + y_j) = \sum_i p_i u(x_i + \hat{y})$, where independence is captured by the fact that the probabilities $q$ do not depend on index $i$, and the exponential property was used again. Thus $CE' = CE^{\dagger}$ and $VoH = CE' - CE^{\dagger} = 0$.    □

**Proof of Theorem 9(i)**:

*Proof.* Setting $u(v) = v$, the difference in certain equivalents is a difference in expected values:

$$\begin{aligned}
VoPH^{\Pi}(X) &= E_{\boldsymbol{\mathcal{X}},\Pi}[f(\boldsymbol{\mathcal{V}}, V')] - E_{\boldsymbol{\mathcal{X}},\Pi}[f(\boldsymbol{\mathcal{V}})] \\
&= E_{\boldsymbol{\mathcal{X}},\Pi}[f(\boldsymbol{\mathcal{V}})] + E_{\boldsymbol{\mathcal{X}},\Pi}[V'] - E_{\boldsymbol{\mathcal{X}},\Pi}[f(\boldsymbol{\mathcal{V}})] \\
&= E_{\boldsymbol{\mathcal{X}},\Pi}[V'] \\
&= 0
\end{aligned}$$

The last step holds because the augmented reward is balanced. The result also holds for $VoPH^*$ after recognizing that the optimal policy does not change here, because the uncertainty $X$ is not downstream of a decision.    □

**Proof of Theorem 9(ii)**:

*Proof.* This result holds for all utility functions. It follows from recognizing that the augmented reward $r(\cdot)$ that yields 0 for all states of $X$ lies within the set of balanced rewards over which the optimization is conducted. In this case, the certain equivalent with the reward is identical to that without it, because the reward outcomes are identical. This ensures a gain over the prior certain equivalent of at least 0, because the optimal balanced reward can only increase $VoPH$ further. Thus $VoPH \geq 0$ (at policy and otherwise).    □

**Proof of Theorem 9(iii)**:

*Proof.* We drop the policy superscript for notational convenience. Consider the augmented reward $r'(X)$ which results in a new value node $V'$. We denote the expected value of the new reward at policy as $\bar{y} = E_{\boldsymbol{\mathcal{X}},\Pi}[V']$. Construct a new reward $r''(X)$ that subtracts $\bar{y}$ from $r'(X)$. This new reward is balanced for additive value aggregation.

Let us denote the certain equivalents with the original augmented reward and the modified balanced reward as $CE_a$ and $CE_m$ respectively. Since the decision maker has an exponential utility function, these certain equivalents are related as follows: $CE_a = CE_m + \bar{y}$. From the definition of a perfect hedge, $CE_m - CE_0$ is bounded above by $VoPH(X)$ (since this balanced hedge might be improved by the perfect one), therefore $CE_a$ - $CE_0$ is bounded above by $VoPH(X) + \bar{y}$. $\qquad\qquad\square$

**Proof of Theorem 10**:

*Proof.* We drop the policy superscript for notational convenience. We denote the perfect hedge (augmented reward) on the state $x$ of uncertainty $X$ as $r^*(x)$. To find this perfect hedge on $X$ (at policy), we note that maximizing the certain equivalent is the same as maximizing the expected utility in the first term of the definition of $VoPH^\Pi$. A non-linear optimization problem can be set up where the objective function is the expected utility with an augmented reward, and the constraint is the balanced hedge requirement for additive value aggregation [4]:

$$\max_{r'(x)} \left(E_{\boldsymbol{\mathcal{X}},\Pi}[f(\boldsymbol{\mathcal{V}}, V')]\right) \text{ s.t. } E[r'(x)] = 0$$

For a risk-averse exponential utility function, the objective function is concave and the constraint is linear, therefore the necessary conditions are also sufficient. To solve the optimization problem, we replace the objective with the specific exponential utility function form $u(v) = a - exp(-v/\rho)$, and treat uncertainty $X$ separately from others in the model.

Solving the constrained non-linear optimization problem using a Lagrange multiplier, the optimal hedge is obtained as: $r^*(x) = E_X(CE|X) - CE(x)$, where $E_X(CE|X)$ is the expected value over the certain equivalents where $X$ is observed to be in state $x$. This is an intuitive result: the perfect hedge across all states $x$ of $X$ is one that considers all possible conditional certain equivalents, i.e. where $x$ has been observed, and brings them all to the same level – a probability weighted average of these conditional certain equivalents, $E_X(CE|X)$. Replacing $r^*(x)$ in the objective function to compute the maximum expected utility and then the maximum certain equivalent, though a logarithmic operation, returns objective of $E_X(CE|X)$. Thus, $VoPH(X) = E_X(CE|X) - CE_0$. $\qquad\square$

# B  Computational Details for the Examples

## B.1  The Game

### B.1.1  Flipped Coin Toss Version

In this version of the game, the (monotonically increasing) exponential utility function for both players follows $a - b\,\text{sgn}(\rho)\exp(-x/\rho)$, where $a$ and $b > 0$ are constants and $\text{sgn}(\rho)$ is the sign of the risk tolerance $\rho \neq \infty$. Choosing $a = 1$ and $b = 1$ for simplicity, and since $\rho > 0$, both players have utility function $u(\cdot)$ as $u = 1 - \exp(-v/\rho)$ and inverse utility function $u^{-1}(\cdot)$ as $v = -\rho\log(1-u)$.

For player #1, $CE_0 = u^{-1}(0.5 * (0.9u(1000) + 0.1u(-1000)) + 0.5 * (0.3u(1000) + 0.7u(-1000))) = \$35.72$, after applying the exponential utility function (with $\rho = 3000$) and its inverse. Similarly, for player #2, $CE_0 = u^{-1}(0.5 * (0.9u(-1000) + 0.1u(1000)) + 0.5 * (0.3u(-1000) + 0.7u(1000))) = \$-350.63$.

For computing the $VoH$ from Definition 4, we note that the third time is identical to the first term ($CE_0$), because the augmented reward is just like the original deal, i.e. arises from a single coin toss. For the first term, note that both players are guaranteed to play both games. For player # 1, the potential rewards are \$2000, \$0 and \$ − 2000, with probabilities 0.27, 0.66 and 0.07 respectively. The first term is therefore $u^{-1}(0.27u(2000) + 0.66u(0) + 0.07u(-2000)) = 201.74$. Therefore, for

player #1, $VoH = \$(201.74 - 35.72 - 35.72) = \$130.3$. These computations can be performed similarly for player #2 but with potential rewards $\$ - 2000$, $\$0$ and $\$2000$, with probabilities 0.27, 0.66 and 0.07 respectively. For this player, $VoH = \$(-601.04 + 350.63 + 350.63) = \$100.22$.

### B.1.2 Independent Coin Toss Version

In this version, two games are played where the second coin toss is independent of the first toss. Due to the constant risk aversion property of the exponential utility function, the $CE$ for the combined reward in this case turns out to be twice the $CE$ for each individual coin toss. This is however not true for the case of the logarithmic utility function.

Player #1 has a probability of 0.6 of receiving $\$1000$ from a single toss, if fencing is played and they win, with a probability of 0.45, or if golf is played and they win, with a probability of 0.15. Alternately, their reward is $-\$1000$ with probability 0.4. The first term in the $VoH$ calculation when two independent games are played is $u^{-1}(0.36u(2000) + 0.48u(0) + 0.16u(-2000))$. For an exponential utility function with $\rho = 3000$, this equals 71.44. Therefore, for player #1, $VoH = \$(71.44 - 35.72 - 35.72) = \$0$. Similarly, for player #2, $VoH = \$(-706.26 - 350.63 - 350.63) = \$0$.

The computations differ when player # 1 has a logarithmic utility function $u(\cdot)$ where $u = log(v + w)$ with $w$ as their initial wealth, assumed to be $\$5000$. The inverse utility function $u^{-1}(\cdot)$ is $v = \exp(u) - w$. In this case, it can be shown that $CE_0 = 101.7$ and the first term in the $VoH$ definition is 283.18, thus $VoH = \$(283.18 - 101.7 - 101.7) = \$79.78$.

### B.2 The Oil Wildcatter

Suppose all variables in Figure 2(a) are binary: the decision is drill ($d$) or not ($n$) and uncertainties are either high ($h$) or low ($l$). Consider the following numbers, where values are in million dollars: cost of drilling = 20 and revenues from drilling are $v(a = h, op = h) = 320$, $v(a = h, op = l) = 140$, $v(a = l, op = h) = 70$, $v(a = l, op = l) = -30$. The decision maker has an exponential utility function with risk tolerance $\rho = 500$. Probabilities are as follows: $P(a = h) = 0.3$, $P(op = h) = 0.4$, $P(ap = h|op = h) = 0.2$, $P(ap = h|op = l) = 0.9$. Note that oil price and airline price show 'negative dependence'.

The decision maker has an exponential utility function with $\rho = 500$, thus we use $u(\cdot)$ as $u = 1 - \exp(-v/\rho)$ and inverse utility function $u^{-1}(\cdot)$ as $v = -\rho \log(1 - u)$. For the drilling alternative ($d$), $CE_0^d = u^{-1}(0.12u(300) + 0.18u(120) + 0.28u(50) + 0.42u(-50)) = \$38.99$ million. For the not drilling alternative ($n$), $CE_0^n = u^{-1}(u(0)) = \$0$, therefore drilling is profitable and therefore preferred a-priori.

$VoPH$ computations can be performed by solving the optimization problem in Definition 7, but since the policy is fixed here (at alternative $d$), we can leverage the result from Theorem 10 for an easier and more efficient computation. Let us demonstrate this for perfect hedging on the amount of oil (A) in the reservoir, which has states high (h) and low (l). Conditioning on the states of the amount of oil, $CE^d|h = u^{-1}(0.4u(300) + 0.6u(120)) = \$184.45$ million and $CE^d|l = u^{-1}(0.4u(50) + 0.6u(-50)) = -\$12.36$ million. Therefore $VoPH^d(A) = (0.3CE^d|h+0.7CE^d|l)-CE_0^d = \$7.68$ million. Similar computations can be performed for perfect hedging on the other two uncertainties - oil price (O.P.) and airline price (A.P.). This example is based on prior work [39].

## C  Experimental Details

We provide further experimental details in this Appendix.

### C.1  Gold Mining

For each of the three stocks, $j \in \{\text{ABX}, \text{BGO}, \text{AEM}\}$, $j$'s monthly return is modeled through linear regression as:

$$R_j = \beta_0 + \beta_M M + \beta_G G + \epsilon_j,$$

where $M$ and $G$ are the market and gold-industry specific indices, $\{\beta_0, \beta_M, \beta_G\}$ are regression coefficients, and $\epsilon_j$ is a stock-specific Gaussian error term, i.e. $\epsilon_j \sim N(\mu_j, \sigma_j^2)$.

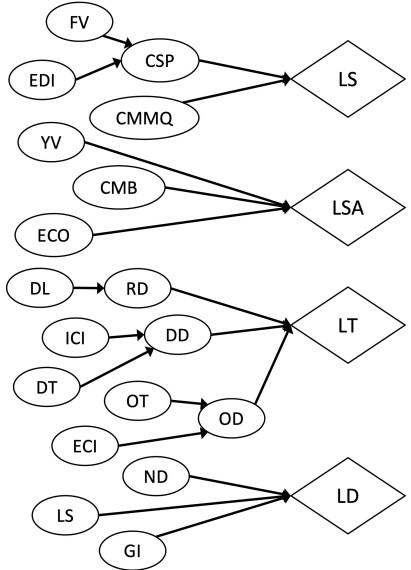

Figure 4: BN for the Supply Chain Risk application: Graphical structure of the Bayesian network.

The supplier condition variable is assumed to be binary with states 'favorable' ($f$) and 'unfavorable' ($u$). The prior probability of 'favorable' condition is $0.7$. Supplier condition is assumed to only affect the mean of the error terms for stocks ABX and AEM, who have the same supplier.

We make the following parameter choices, most of them from real data analyzed by [42]:

- Market index $M \sim N(0.55, 2.28^2)$
- Gold index $G \sim N(-0.23, 1.43^2)$
- ABX
    - $\beta = [\beta_0, \beta_M, \beta_G] = [0.17, 0.37, 2.26]$
    - $\mu_{ABX} = 0$ for state $f$ and $-1$ for $u$; $\sigma_{ABX} = 3.63$
- BGO:
    - $\beta = [\beta_0, \beta_M, \beta_G] = [1.48, 0.26, 3.96]$
    - $\mu_{BGO} = 0$; $\sigma_{BGO} = 11.27$
- AEM
    - $\beta = [\beta_0, \beta_M, \beta_G] = [0.05, 0.27, 2.76]$
    - $\mu_{AEM} = 0$ for state $f$ and $-1$ for $u$; $\sigma_{AEM} = 5.27$

## C.2  Supply Chain Risk

All figures from Figure 4 onward provide the qualitative as well as quantitative details about the Supply Chain Risk Bayesian network application. Our version of the model involves most but not all variables from [9], for simplicity. Most numbers are based on those from data and/or expert assessments as described in [9].

FV – Forecast Variability
EDI – Electronic Data Interchange Outages
CSP – Component Sourcing Problems
CMMQ – Contract Manufacturer Management Qualifications
YV – Yield Variability
CMB – Contract Manufacturer Bankruptcy
ECO – Engineering Change Orders
DL – Damage and Loss
RD – Rebuild Decision
ICI – Import Clearance Time
DT – Destination Trucking
DD – Destination Delay
OT – Origin Trucking
ECI – Export Clearance Time
OD – Origin Delay
ND – Natural Disaster
LS – Labor Strike
GI – Geo-political Instability

LS – Leadtime Sourcing
LSA – Leadtime Sub-Assembly
LT – Leadtime Transportation
LD – Leadtime Disaster

Figure 5: BN for supply chain risk: List of uncertain variables and rewards, with full forms of acronyms.

| EDI | P(EDI) |
| --- | --- |
| yes | 0.04 |
| no | 0.96 |

| CMMQ | P(CMMQ) |
| --- | --- |
| poor | 0.1 |
| good | 0.9 |

| YV | P(YV) |
| --- | --- |
| yes | 0.004 |
| no | 0.996 |

| ECI | OT | OD | P(DD|ICI,DT) |
| --- | --- | --- | --- |
| yes | high | high | 1.0 |
| yes | high | low | 0.0 |
| yes | low | high | 0.5 |
| yes | low | low | 0.5 |
| no | high | high | 1.0 |
| no | high | low | 0.0 |
| no | low | high | 0.3 |
| no | low | low | 0.7 |

| FV | EDI | CSP | P(CSP|FV,EDI) |
| --- | --- | --- | --- |
| under | yes | complex | 0.5 |
| under | yes | simple | 0.5 |
| under | no | complex | 0.1 |
| under | no | simple | 0.9 |
| not_under | yes | complex | 0.2 |
| not_under | yes | simple | 0.8 |
| not_under | no | complex | 0.1 |
| not_under | no | simple | 0.9 |

| ECO | P(ECO) |
| --- | --- |
| yes | 0.05 |
| no | 0.95 |

| CMB | P(CMB) |
| --- | --- |
| yes | 0.02 |
| no | 0.98 |

| DL | P(DL) |
| --- | --- |
| yes | 0.04 |
| no | 0.96 |

| FV | P(FV) |
| --- | --- |
| under | 0.6 |
| not_under | 0.4 |

| DT | P(DT) |
| --- | --- |
| high | 0.2 |
| low | 0.8 |

| ECI | P(ECI) |
| --- | --- |
| yes | 0.005 |
| no | 0.995 |

| OT | P(OT) |
| --- | --- |
| high | 0.1 |
| low | 0.9 |

| ICI | DT | DD | P(DD|ICI,DT) |
| --- | --- | --- | --- |
| yes | high | high | 1.0 |
| yes | high | low | 0.0 |
| yes | low | high | 0.7 |
| yes | low | low | 0.3 |
| no | high | high | 1.0 |
| no | high | low | 0.0 |
| no | low | high | 0.1 |
| no | low | low | 0.9 |

| DL | RD | P(RD|DL) |
| --- | --- | --- |
| yes | yes | 0.2 |
| yes | no | 0.8 |
| no | yes | 0.0 |
| no | no | 1.0 |

| ICI | P(ICI) |
| --- | --- |
| yes | 0.02 |
| no | 0.98 |

| LS | P(LS) |
| --- | --- |
| yes | 0.2 |
| no | 0.8 |

| GI | P(GI) |
| --- | --- |
| yes | 0.02 |
| no | 0.98 |

| ND | P(ND) |
| --- | --- |
| yes | 0.1 |
| no | 0.9 |

Figure 6: BN for supply chain risk: Conditional probability tables (CPTs).

| CSP | CMMQ | LS |
|---|---|---|
| complex | poor | 8 |
| complex | good | 5 |
| simple | poor | 3 |
| simple | good | 2 |

| YV | CMB | ECO | LSA |
|---|---|---|---|
| yes | yes | yes | 21 |
| yes | yes | no | 21 |
| yes | no | yes | 6 |
| yes | no | no | 6 |
| no | yes | yes | 21 |
| no | yes | no | 18 |
| no | no | yes | 6 |
| no | no | no | 3 |

| RD | DD | OD | LT |
|---|---|---|---|
| yes | high | high | 12 |
| yes | high | low | 11 |
| yes | low | high | 11 |
| yes | low | low | 10 |
| no | high | high | 7 |
| no | high | low | 6 |
| no | low | high | 6 |
| no | low | low | 5 |

| ND | LS | GI | LD |
|---|---|---|---|
| yes | yes | yes | 56 |
| yes | yes | no | 29 |
| yes | no | yes | 54 |
| yes | no | no | 27 |
| no | yes | yes | 29 |
| no | yes | no | 2 |
| no | no | yes | 27 |
| no | no | no | 0 |

Figure 7: BN for supply chain risk: Reward functions.