# OpenReview forum: "Hedging as Reward Augmentation in Probabilistic Graphical Models"
_NeurIPS.cc/2022/Conference — NeurIPS 2022 Accept_

### Official Review · Reviewer_XnHK · 2022-07-09

**Rating:** 5
**Confidence:** 3
**Soundness:** 3 good
**Presentation:** 3 good
**Contribution:** 2 fair

**Summary:**

The authors propose a modified Bayesian Network formalism incorporating the hedging information that can be useful in the decision making process.

**Questions:**

Fig. 1 - Some variables are not defined. This is a problem present in many figure where the caption is not so informative.

row 90 --> what is the meaning of "numeraire"?

row 122  --> $\mathbf{V_k}$ should be substituted by $V_k$ since we refer to a particular Value node.

row 131 --> what is the alternative set?


**Limitations:**

There isn't a mention neither to the Markov Decision Process nor to the Control as Probabilistic Inference (I will not make a list of papers here to avoid favoritism but a simple search for "Control as Probabilistic Inference" should give the authors plenty of references).
The difference with these approaches should be described because it is not clear what is the advantage of the proposed approach.

The Reinforcement Learning (RL) literature on this topic is not considered (e.g. risk-aversion RL). RL is cited only regarding the risk in the reward manipulation.

**Strengths And Weaknesses:**

The main weakness I see is an insufficient contextualization of the problem. The Markov Decision Process, the control as probabilistic inference, the risk-aversion Reinforcement Learning, etc. are neither mentioned nor discussed.

There isn't presented any alternative approach to solve the problems presented and so any comparison is missing.

---

> ### Author Response · Authors · 2022-08-02
> **Response**
>
> We thank the reviewer for their feedback and suggestions.
>
> **Regarding contextualization of the work w.r.t MDPs/RL:**
>
> We assure the reviewer that we are very familiar with MDPs and RL. The reason they were not highlighted is because they are not directly relevant for our work. Some reasons follow:
>
> - Our work assumes that a probabilistic graphical model is known; this may have been learned through a combination of data and expert domain knowledge. There is no aspect of learning here – only planning under uncertainty – and therefore there is no need to refer to RL explicitly.
> - Influence diagrams are graphical representations of decision problems (under uncertainty); they can also represent MDPs/POMDPs. They are often used for problems with some of the following characteristics: 1) short episodes (sometimes even only 1 or 2 decisions); 2) different types of variables are observed before different decisions, unlike in typical MDPs where state variables remain the same, 3) it is suitable to model several key uncertainties and their probabilistic relationships. We note however that our work extends to typical MDP problems since influence diagrams are very general representations.
> - Our contributions are conceptual, and our work is agnostic to how an ID is solved or how inference is conducted in a BN. One can use any solution approach, including techniques popular for solving an MDP, if that is what the ID represents.
> - Hedging as we have defined it can also be computed for BNs (with value/reward nodes), not just for problems with decisions such as MDPs.
> - The idea of relating decision making (by maximizing expected utility) with probabilistic inference is well known in the graphical model community, at least since the 1990s.
>
> We urge the reviewer to rethink their perspective on the paper, given the above comments, since they do not appear to be familiar with probabilistic graphical models. If they still feel that making a connection to MDPs would be helpful for the generic reader, we can provide more context and relate the work to some of the MDP/RL literature.
>
>
> **Responses to other comments:**
>
> >> Fig. 1 - Some variables are not defined. This is a problem present in many figures where the caption is not so informative.
>
> In Fig. 1, only variables X and D are truly pertinent, and they were both mentioned in the caption. There is not enough space to define all variables in the captions of figures since some of the examples have many variables. In our opinion, these are not difficult to decipher from the main text or supplemental material, but we will check again and make some captions more informative.
>
> >> row 90 --> what is the meaning of "numeraire"?
>
> We use “numeraire” just to specify that this is a real-valued number, but we will leave it out since it is confusing and not needed.
>
> >> row 122 --> \mathbf{V}_k should be substituted by V_k since we refer to a particular Value node.
>
> We thank the reviewer for catching this typo.
>
> >> row 131 --> what is the alternative set?
>
> The alternative set is the set of alternatives/actions of a decision.

---

> > ### Comment · Reviewer_XnHK · 2022-08-04
> > **Response**
> >
> > I thank the authors for their answers
> >
> > >We assure the reviewer that we are very familiar with MDPs and RL.
> >
> > Nobody tells differently and surely I only described the limitations I saw in your work. Only your work is the subject of *my* opinions.
> >
> > The clarifications you made regarding your view on contextualization w.r.t. MDPs/RL could be useful for a generic reader.
> > Probably I didn't make my point clear.  My main concern is that the risk aversion is present in context that are similar to yours (e.g. search risk aversion in MDP and obtain several works on it). I understood that your approach is more general but you should make this point clearer. A reader could think that the concept of risk aversion is already present, in different form, in other works and, for this reason, I think that a clear discussion on difference/originality of your work is worth to do (or at least explain clearly why you think that literature is not relevant to your work).
> >
> > I know that decision making as probabilistic inference is not new but my point remains, the introduction of optimality variables in a bayesian network sounds me related to your idea.
> >
> > >We urge the reviewer to rethink their perspective on the paper, given the above comments, since they do not appear to be familiar with probabilistic graphical models. If they still feel that making a connection to MDPs would be helpful for the generic reader, we can provide more context and relate the work to some of the MDP/RL literature.
> >
> > Even though authors think I'm not familiar with PGMs, I understood quite well that the contribution is conceptual and general, but I pose myself as a generic reader and try to give suggestions and opinions as required to reviewers.
> > Anyway, reading again the paper, looking other reviewers' comments, and the authors' responses, gave me more elements and I decided to modify my initial rating.

---

> > > ### Author Response · Authors · 2022-08-04
> > > **Follow-up**
> > >
> > > We thank the reviewer for further clarifications as well as for spending the time to peruse other reviews and responses.
> > >
> > > We understand their point better about mentioning related work on risk-aversion in MDPs and will briefly refer to this work in the literature review. We will also try to bring in some of the content from this discussion to provide context to a generic reader regarding how influence diagrams are general graphical representations of sequential decision making problems.

---

### Official Review · Reviewer_GBff · 2022-07-10

**Rating:** 6
**Confidence:** 4
**Soundness:** 4 excellent
**Presentation:** 4 excellent
**Contribution:** 2 fair

**Summary:**

This paper studies hedging within the framework of causal influence diagrams (CID). Roughly speaking, a hedge is an opportunity for a risk-averse agent to increase its expected utility, by creating anti-correlated sources of value. This paper formalises several concepts relevant to hedging within the CID-framework, and proves a number of theorems that follow from these definitions. These theorems can then be used to identify potential hedges in an uncertain decision situation, or reason about the incentives of an agent. This paper builds on recent work on CIDs, as developed by Tom Everitt and others.

**Questions:**

I have the following questions:
- What is the main (practical or theoretical) significance of the results presented in the paper?
- What problems can the presented results help us to solve, or gain more insight into (as compared to a naive analysis)?
- What further theoretical results or extensions do you anticipate that it will be possible to build on the results you have provided?
- Is there a theoretical reason to expect the provided results to be exhaustive (in some relevant sense)?

I believe that a better understanding of these points could compel me to increase my rating.


**Limitations:**

I find that the authors have addressed the potential negative societal impact of their work in a fully satisfactory way. Their discussion of the limitations of their work is also adequate (although dispersed throughout the text).

**Strengths And Weaknesses:**

Originality:

There is, to the best of my knowledge, no previous work studying hedging within a similar framework.


Quality:

The presented results are of good quality. The maths is sound, and the experiments serve their purpose.


Clarity:

I found the presentation and exposition in this paper to be exceptionally clear.


Significance:

My impression is that the contribution made by this paper is somewhat marginal. It studies a relevant problem from a promising direction, but the results themselves could be more substantial. The first two theorems (Theorem 6 and 9) are fairly straightforward. Theorem 10 seems to be more substantive, but its implications are not discussed in the paper in any great detail. Of course, for a paper like this, the definitions and framework constitute a contribution by themselves, but it is not clear to me that this is enough to make the paper impactful in this case. I would have liked to see a more complete treatment of the problem, either with more theoretical results, or with case-studies that demonstrate how these results could help us to solve or better understand some class of problems (be they practical or theoretical). Alternatively, it would also suffice to provide a compelling argument for why such extensions would be very difficult to provide (thus arguing that there is a sense in which the provided results are complete).

---

> ### Author Response · Authors · 2022-08-02
> **Response**
>
> We thank the reviewer for valuable feedback and will respond below to specific questions. We have re-ordered the questions slightly just to provide more flow to our response.
>
> We first make a clarification: the concepts introduced here apply to any BN/ID, making them extremely general. Thus, they do not apply to only IDs where there are additional causal semantics. BNs and IDs have been around for decades. Our work here provides a clearly justified tool for risk averse decision makers to analyze their BN/ID.
>
> >> What problems can the presented results help us to solve, or gain more insight into (as compared to a naive analysis)?
>
> The presented results apply to any situation that can be represented as a BN/ID. This is a broad category of decision situations where the model captures probabilistic relationships between potentially several uncertainties. Our results apply to models of any size, i.e., number of nodes/edges in the underlying graphical representation, as long as they can be solved with state-of-the-art algorithms. The supply chain risk example in Section 5.3 is illustrative of the sort of problem where our techniques could be beneficial. Decades of research on formulating and learning BNs/IDs from data and/or by eliciting information from experts precede our work, and all of those motivating problems are candidates for utilizing our results.
>
> The main advantage as opposed to naïve approaches (such as looking solely at anti-correlation measures) is a formal and well-justified definition/framework for hedging in probabilistic graphical models that incorporates all aspects of the problem that are important: the model of relationships between uncertainties, the decisions (if present), values/rewards involved, and the decision maker’s risk preference.
>
> >> What is the main (practical or theoretical) significance of the results presented in the paper?
>
> The significance is three-fold: a conceptual framework for hedging, theoretical results for various important practical cases, and a practically useful analysis tool for risk-averse users of probabilistic graphical models. We briefly touch upon each aspect below.
>
> The reviewer alluded to this point – the primary significance is indeed with regard to the general conceptual definitions and framework for hedging in probabilistic graphical models. We have shown hedging to be a graph manipulation, analogous to information and control, which have been studied in the literature for decades. Importantly, the definitions were justified using value increments, in analogy to value of information, which is well studied across domains including AI/ML.
>
> While some of the theoretical results are not difficult to derive, they can be beneficial for users at a high level of generality. For instance, a risk-neutral decision maker would not need to worry about hedging at all, regardless of their model, according to our framework. Sometimes the results conform to our expectations, such as in Th 9 (i) and (ii), while at other times, they can be counterintuitive, as we have shown using examples. The theoretical results therefore provide some insight into how hedging computations are affected by the various inputs, wherever possible.
>
> From a practical standpoint, we refer the reviewer back to the supply chain risk example from Section 6.3, which is illustrative of how the proposed framework could be used as an additional tool to analyze BNs/IDs. Even if theoretical results don’t apply, the computations can be performed as proposed here, and utilized to identify potential hedging opportunities.
>
> >> What further theoretical results or extensions do you anticipate that it will be possible to build on the results you have provided? Is there a theoretical reason to expect the provided results to be exhaustive (in some relevant sense)?
>
> We respond to these two questions together as they are closely related. Our results are reasonably complete for how we have set up the framework. It is of course possible for others to provide their own definitions of hedging in probabilistic graphical models, but we have argued that our definitions make conceptual sense and are analogous to prior well-studied related ideas. While there might be theoretical extensions for special cases, we feel extensions are more likely to be algorithmic in nature. For instance, there is an open question on how to compute Equation (4) efficiently, as we mention in Section 6.
>
> Some of the theoretical results are indeed exhaustive, due to the properties of some utility functions. For instance, only the linear and exponential utility functions are invariant to shifts (this is a well-known theoretical result), and therefore it will be impossible to extend some results to other utility functions for additive aggregation of value.
>
>
> We hope our responses have shed some more light on the benefits (theoretical and practical) of the proposed techniques, and that the reviewer will consider increasing their rating.

---

### Official Review · Reviewer_fGwk · 2022-07-12

**Rating:** 7
**Confidence:** 3
**Soundness:** 3 good
**Presentation:** 4 excellent
**Contribution:** 3 good

**Summary:**

This paper introduces a new method for studying the value of hedging on uncertain variables in decision-making problems. This is done by augmenting the reward with the value of the hedge. That way, gains/losses associated with a particular hedge can be computed so that alternative scenarios such as no hedging or hedging on another uncertainty can be evaluated and compared.

**Questions:**

Aside from addressing the concerns listed under weaknesses I would like the authors to answer the following questions:

1. As mentioned under weaknesses, a proper description of certain financial concepts is much needed. For instance:

     * How should the utility function be chosen? Is this given? How do the different utility functions affect the final policy/decision? To be more specific, I found Section 3.1 hard to follow. I suggest the authors expand it with examples or intuitions illustrating these concepts.

     * The notion of certain equivalents is also unclear. Can the authors give an intuition on why it is the inverse of the utility function?

     * The last sentence of the second paragraph in Section 4 is confusing: "The agent’s certain equivalent (CE) for their (decision) situation (represented by the ID/BN) is the certain amount, in units of value V, at which they are indifferent between this amount and their situation."

2. The paper discusses how to compute VoPH for a particular policy when the rewards are unknown (Theorem 10). However, equation 4 includes an inner maximization to find the optimal policy, which itself depends on the augmented reward. Can the authors explain how equation 4 can be solved?



**Strengths And Weaknesses:**

### Strengths
* The paper is clearly written and gives many examples illustrating the method and showing the different cases that may occur: hedging for a fixed policy, hedging with policy re-optimization.

* It also provides theoretical results showing how the calculations for the VoPH when the rewards are unknown can be done efficiently using graphical models and BN inference methods.

Overall, I think this is a strong paper that will be valuable for the community. A list of my main concerns is provided below. I would like the authors to comment on them in their rebuttal and revise the paper if needed.

### Weaknesses

* The paper seems to assume that the reader is familiar with concepts from finance such as hedging, risk aversion, or utility functions. Given that this is an ML conference it would be good to introduce all these more carefully (see my comment under Questions for more details). The authors may save some space by shortening section 3.2 introducing BNs and IDs which should be familiar concepts for the ML audience.

* The terminology is confusing at times, the paper uses the terms reward, value and utility interchangeably.

* There is a big gap in the related work section. The paper ignores all the works on reward shaping in RL. I will not make a list of papers here to avoid favoritism but a simple search for "reward shaping in RL" should give the authors plenty of references. Although the objectives in most of these papers (e.g. encourage exploration (intrinsic motivation) or ease learning) are different in nature from the objective in this paper (studying the value of hedging), the means to accomplish these goals are similar. In fact, it is difficult to assess how different this paper is from all these works without the proper context.

* The paper also fails to discuss the assumptions and limitations of the method. Such as the availability of a graphical model of the problem. Full knowledge of the dynamics is a strong assumption that should be mentioned in the paper.

---

> ### Author Response · Authors · 2022-08-02
> **Response**
>
> We thank the reviewer for valuable feedback and constructive comments that will improve the paper. Some specific responses follow:
>
> >> “The paper seems to assume that the reader is familiar with concepts from finance such as hedging, risk aversion, or utility functions …”
>
> We will do our best to add further detail (such as examples) about key concepts, specifically, risk aversion, utility functions and certain equivalent. Our challenge is that since this work is interdisciplinary, we need to describe many different ideas briefly. For instance, one of the reviewers is clearly quite unfamiliar with BNs/IDs, and we feel this has strongly affected their evaluation of the work. We will however expand upon the content in Section 3.1 as the reviewer suggests in a separate comment.
>
> >> “ … the paper uses the terms reward, value and utility interchangeably”
>
> We were careful to maintain utility as an entirely separate concept, but we tried to be consistent in using reward and value interchangeably – indeed, at various places, we use both terms together like value/reward. We will make this explicit in Section 3.1.
>
> >> “The paper ignores all the works on reward shaping …”
>
> We agree with the reviewer that any work on affecting rewards is relevant literature to cite. We have cited some work pertaining to reward hacking, but we agree that some mention of reward shaping will be beneficial as well. We will add some references at the end of Section 2. We do not however think that reward shaping is directly relevant to what we are trying to achieve, because the goal in reward shaping is to make it easier to learn using supplemental rewards. In our proposal for hedging, there is no learning component – only an attempt to explore opportunities for minimizing risk as suitable for the decision maker.
>
> >> “The paper also fails to discuss the assumptions and limitations of the method. Such as the availability of a graphical model of the problem. Full knowledge of the dynamics is a strong assumption that should be mentioned in the paper.”
>
> We feel we made it clear that the model is assumed to be known, for instance, please see the first contribution in Section 1 (“ … demonstrate how to evaluate hedging opportunities given a model”). However, we will make it even clearer and mention it as a potential limitation. The potential advantage of course is that our proposed analyses techniques are agnostic to how models are learned/elicited.
>
>
> **Responses to Questions:**
>
> *Choice of utility function:*
>
> Yes, we assume that this is specified by the decision maker. This reflects the decision maker’s risk preference and is subjective. However, in various places in the paper, we explore sensitivity to a utility function parameter to shed some light on its effect. It definitely affects hedging valuations, as one would expect (and desire, in our view).
>
> *Certain equivalent (CE):*
>
> This is an old idea that goes back all the way to at least Laplace. A common definition for CE is the selling price of an uncertain deal for a decision maker. Thus in our work, making a simplification by ignoring a decision maker’s wealth, if a decision maker has a deal with 2 outcomes with utilities u1=u(v1) and u2 = u(v2) with probabilities p and 1-p respectively, then the following must hold: u(CE) = p * u1 + (1-p) * u2, therefore CE = u^{-1}( p * u1 + (1-p) * u2). We have equated two expected utilities from two different situations – one where the decision maker has a fixed certain CE (in units of value/reward), and another where they have the uncertain deal. This is where the inverse comes in and what we meant by the sentence in the second para of Section 4.  Note that the decision maker’s risk preference matters through their utility function u(.). We will clarify this in the paper, perhaps with the above example.
>
> *Inner maximization:*
>
> The inner maximization actually goes away for this specific result since it applies only for a fixed policy \Pi (Equation (3) and not Equation (4)). The decision maker can fix a policy, perhaps the one that optimizes the original problem, and then compute VoPH^{\Pi}(X). When one considers potential re-optimization as in Equation (4) to compute VoPH^{*}(X), this becomes a harder computation in general, so approximate methods may be required. We mention this in the last line of Section 5 and also in Section 6 as a potential direction for future work.

---

> > ### Comment · Reviewer_fGwk · 2022-08-05
> > **Reviewer response**
> >
> > I would like to thank the authors for their detailed response, which has helped clarify my questions.
> >
> > Do I understand correctly that the CE is equivalent is just the expected value of a situation? If so, why don't you just write $E_x[\mathcal{V}]$ in Definition 1? Is it because the expectation can not be moved outside $u^{-1}$ for some reason?
> >
> > I would have preferred that the authors actually implemented the changes in the revision rather than promising they will do so if the paper is accepted.

---

> > > ### Author Response · Authors · 2022-08-05
> > > **Followup**
> > >
> > > The CE is not the expected value of the reward in general, except for the case when the decision maker is risk-neutral. Note that in the example provided, CE = u^{-1}(p * u1 + (1-p) * u2). It only equals the expected reward of EV = p * v1 + (1-p) * v2, when u(v) = v (risk-neutral). For a risk-averse decision maker, CE is less than the EV.
> > >
> > > We have been unable to make edits due to a combination of travel and illness (catching Covid). We will however be sure to incorporate the reviewers' helpful suggestions.

---

### Meta-Review · Area_Chair_6FGq · 2022-08-26

**Recommendation:** Accept
**Confidence:** Less certain

**Metareview:**

This paper proposes a decision-theoretic view of hedging within the framework of probabilistic graphical models augmented with a reward.
After reading each other's reviews and the authors' feedback, the reviewers have solved most of their concerns and agree that the paper deserves publication.
However, the authors need to seriously consider the reviewers' suggestions for making their paper clearer in the camera-ready version.

**Award:**

No

---

### Decision · Program_Chairs · 2022-09-14

Accept